

# Development of an Expendable Current Profiler Based on Modulation and Demodulation

Keyu Zhou [1], Qisheng Zhang [1,*], Guangyuan Chen [2], Zucan Lin [1], Yunliang Liu [2] and Pengyu Li [1]

[1]School of Geophysics and Information Technology, China University of Geosciences (Beijing), Beijing 100083, China
[2]School of Marine Science and Engineering, Shandong University of Science and Technology, Qingdao 266000, China

*Correspondence to: zqs@cugb.edu.cn

**Abstract.** We designed a low-cost expandable current profiler including software and hardware. An expendable current profiler (XCP) is an observation instrument that rapidly measures currents based on the principle that currents cut the

geomagnetic field to induce electric fields. It is important to reduce the cost of an XCP because it is a single-use device. The digitization of the previously developed XCP is carried out underwater, which requires the probe to contain not only analogue circuits for acquiring signals but also digital circuits and digital chips, which are relatively expensive. In this study, an XCP has been developed that adopts signal modulation and demodulation to transmit analogue signals on an enamelled wire, and the signal digitization occurs above the surface of the water. The cost of the instrument is effectively reduced by half while

maintaining the ability to measure parameters such as sea current and temperature in real-time. After comparison with data processed from laboratory tests, the acquisition circuit showed accuracy within one-thousandth of one per cent, and the XCP analogue circuit developed for the overall system was stable and reliable. The system exhibited an acquisition accuracy higher than 50 nV for 16 Hz, and the quality of the acquired signal met the requirements for an XCP instrument.

## 1. Introduction

The expendable current profiler (XCP) is a single-use instrument that rapidly measures currents, mainly the velocity and flow direction of seawater, based on the principle that currents cut the geomagnetic field, inducing electromagnetic fields (Liu, 2017). The commonly used instrumentation methods for the measurement of ocean currents can be classified into floating, electromagnetic, mechanical, acoustic, and other (Peltier, 2013; Simpson, 2001; Jenkins, 2006; Le Menn, 2020). XCP is an electromagnetic current meter that produces rapid measurements using geomagnetism and uses a non-stop and non-recovery

mode of operation, offering the advantages of a short detection period, instantaneous data acquisition, wider detection range, and various deployment forms (Niiler, 1991; Zhang, 2014; Hibiya, 2006; Jichang, 2009). Because XCP is a single-use device and will not be recycled, it is important to reduce its cost.

Sanford *et al.* proposed a basic formula for the calculation of the electromagnetic field induced by seawater motion in 1971 based on Faraday's law of electromagnetic induction (Dunlap, 1981; Sanford, 1982). They then developed the prototype of

XCP in 1978 and conducted joint sea trials with the company Sippican, yielding preliminary test results (Dunlap, 1981; Sanford, 1982). At present, the core detection technology remains owned by the American and Japanese companies Sippican and

Tsurumi−Seiki, respectively, and is embargoed in some countries (Liu, 2017, Szuts, 2012). In addition, the instruments of the companies have not been enhanced or improved since 2005. More recently, some Chinese universities and research institutes have started to research and develop XCPs. The authors successfully developed an XCP for digitalization underwater (Liu, 2017), whereby the acquired analogue signal is digitized and then transmitted through a double-strand enamelled wire with an outer diameter of 0.1 mm. Testing showed that the method can transmit more than 1000 m at a transmission band rate of 4800 bps (Liu, 2017; Zhang, 2013; Li, 2018; Liu, 2017; Li, 2017). Digitization underwater requires the probe to contain not only analogue circuits for acquiring signals but also digital circuits and chips, such as the master control chip and related power conversion modules. Furthermore, the buoy board at the surface of the water that forwards and releases the probe uses master control chips, which are relatively expensive. Therefore, the total cost of the XCP is relatively high, which is not suitable for mass production.

Based on this, we have developed an XCP that adopts signal modulation and demodulation to transmit analogue signals on an enamelled wire and performs digitalization above the surface of the water. Since the digital circuitry inside the underwater probe has been removed from our design, the cost of the instrument is effectively reduced while still retaining measurement functionality in real-time on parameters such as sea current and temperature. This should pave the way for widespread adoption and mass production. We also carried out the laboratory analogue circuit test. Through the data processing and comparison of the laboratory test, it was found that the accuracy of the acquisition circuit was within one thousandth of one per cent. The XCP analogue circuit developed by the overall system was stable and reliable. For the 16 Hz signal acquisition, the accuracy was better than 50 nV, and the quality of the acquisition signal met the requirements of the XCP instrument. In addition, we have constructed a laboratory flume simulation environment, tested the biogenic electric field in the simulated sea environment, and compared the results with those acquired by the previously developed XCP, which were found to be highly consistent.

## 2. Entire structure of the XCP

The basic principle of XCP was proposed by Sanford *et al.* in the 20th century (Sanford, 1978; Sanford, 1971) and subsequently analysed and studied by various researchers (Jenkins, 2006; Niiler, 1991; Zhang, 2017), but this will not be further elaborated on here.

The usage procedure of the XCP developed in this study is similar to existing ones. Figure 1 shows the schematic diagram of XCP operations. After the device is cast into the sea, the buoy board is automatically powered upon touching the seawater, and XCP probes are automatically released at regular intervals. The probes rotate and sink at a frequency of 16 Hz and will also be automatically powered up when launched into the seawater. While sinking, the electric field and temperature sensors collect data that are transmitted to the buoy board in the form of analogue signals through an enamelled wire. After the buoy board has demodulated the signals, they are digitized, and the processed data are sent to the host computer via a wireless module. The data is then analysed and processed by the host computer, and the waveform is displayed in real-time.





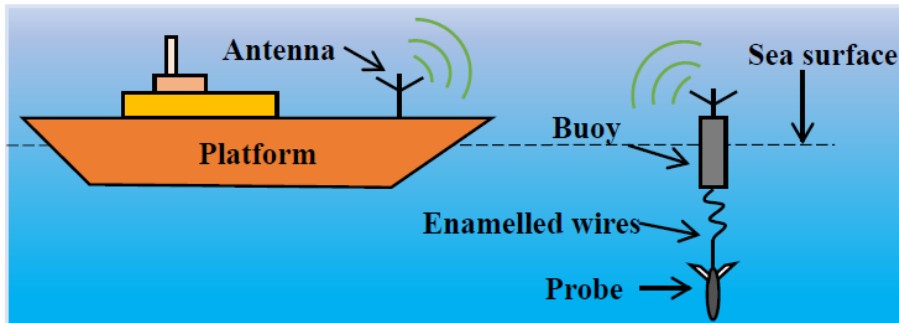

**Figure 1. Schematic diagram of expendable current profiler (XCP) operations.**

The overall architecture of the instrument is shown in Figure 2. The interior of the probe consists of three sensors and an analogue board. The buoy board consists of an analogue board, digital board, wireless transmitter board, and antenna. The receiver on the deck consists of a wireless module board and antenna. The analogue board and buoy board are connected by 1500 m of enamelled wire, which is used to transmit signals from underwater to the surface. For each part of the circuit board, we use lithium batteries for the power supply. Because each module requires different voltages, we use different power

conversion modules (DC-DC) for voltage conversion. At the same time, to reduce power noise, we use a low dropout linear regulator (LDO). LDO has the advantages of lower noise and lower static current. In the third section, we will introduce each part of XCP separately.



**Figure 2. The overall architecture of the XCP.**





## 3. Design of the XCP

### 3.1 Circuit design of the *underwater* part


The analogue board contains a front-end electric field sensor, direction sensor, and temperature sensor. After the analogue signals acquired by the front-end sensors are conditioned, the three voltage signals are converted into three frequency signals with different intervals using voltage−frequency conversion (VFC), which are then superimposed and modulated into one

signal for transmission.

The electric field and compass coil signals are first amplified by the low-power precise instrumentation amplifier INA128 through the electrode and direction sensor and then sent to the voltage-to-frequency converter chip through a series of amplification and filtering. The INA128 is a general-purpose instrumentation amplifier with high accuracy and low power consumption developed by Texas Instruments and is commonly used in situations requiring high accuracy and circuit stability.

The INA128, as an amplifier, is very easy to debug and operates in the temperature range of -40 to 85 °C, while maintaining low drift characteristics with a maximum temperature drift of only 0.5 µV/°C (INA128, Ren; 2011), making it ideal for the undersea operations of the XCP. The schematic diagram of the coil channel acquisition circuit is shown in Figure 3. With the final addition of a second-order controlled voltage source to the Butterworth low-pass filter circuit, the cut-off frequency is set to 16 Hz to filter out high-frequency noise.

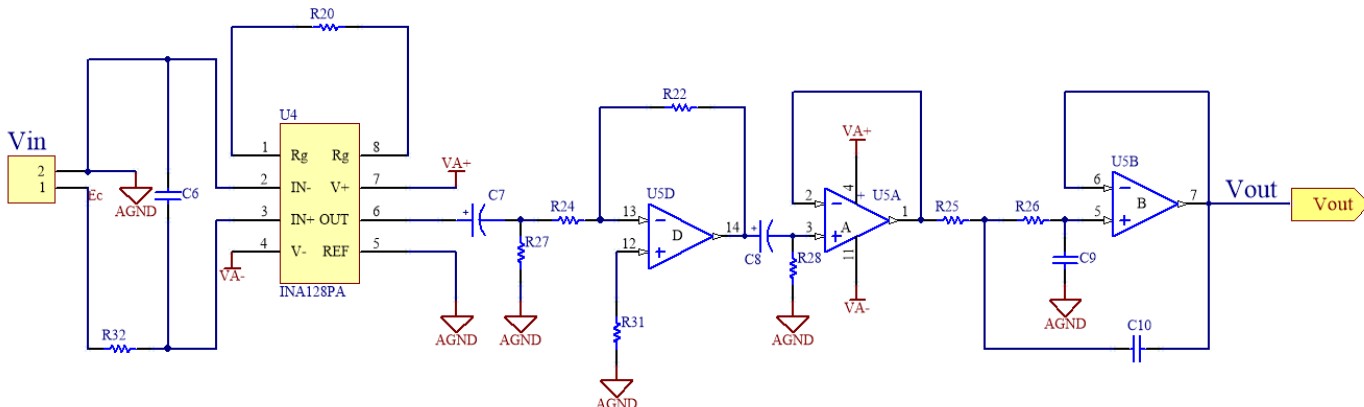


**Figure 3. Schematic diagram of the coil channel acquisition circuit.**

The temperature signal is acquired through a temperature-sensitive resistor using a typical bridge circuit with high-precision voltage dividing resistors. After the temperature information is converted into a voltage signal, it enters the voltage-to-frequency converter chip that then outputs the frequency signal. The voltage-to-frequency conversion circuit modulates the

three signals in different frequency bands using different resistor-capacitor parameters. The three frequency signals are processed by band-pass filters and then superimposed by an adder, modulating them into a frequency division multiplexed signal. This signal is inverted into a signal of the opposite phase but the same amplitude, and then it is connected with the original signal to the two ends of a double-enamelled wire, forming a pair of differential signals for transmission. The





underwater probe section has one less digital board compared to the previously developed XCP (Liu, 2017), which reduces

the cost by approximately half.

Since the acquired signal is extremely weak and of nano voltage level, there are three grounds on the analogue board to reduce interference: analogue, power, and seawater. Connecting each ground with magnetic beads can effectively reduce interference. Figure 4 is the circuit schematic diagram of the switch part of the analogue board that is in contact with water. Power1 is an externally powered lithium battery, with one "key" pin connected to the seawater using the same material as the

electrode, and the other pin connected to the ground of the battery on the board. When the probe enters the seawater, the seawater becomes connected to the board ground, pulling the sixth pin of the CD4013 chip to a low level. This reverses the first-pin level, connects the MOS tube, and supplies the voltage of the lithium battery to the whole board. The water entry power method was tested for speed and stability and could be applied to other expendable instruments.

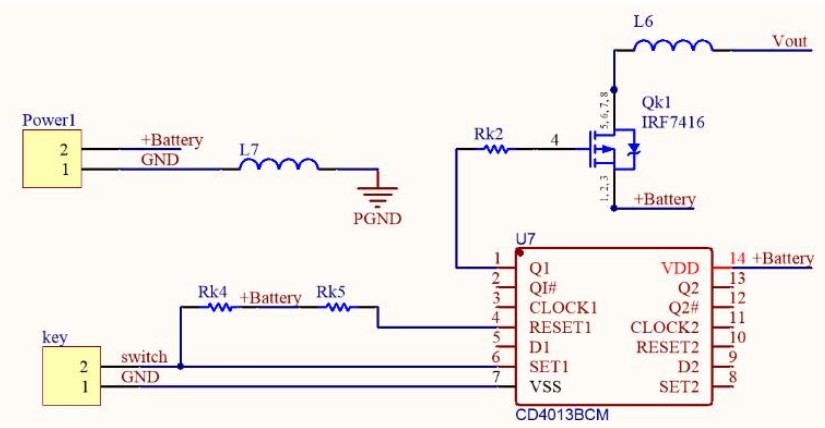

**Figure 4. Circuit schematic diagram of the switch part of the analogue board in contact with water.**

**3.2 Circuit design of above water part**

The circuit board in the buoy is mainly composed of an analogue board, digital board, and wireless transmitter board, and the block diagram of the structure is shown in Figure 5. The analogue board demodulates and conditions the modulated signal on the enamelled wire into a digital signal that can be recognized by the main control chip. First, the differential signal transmitted

through the enamelled wire is isolated and amplified by a transformer. Second, the signal is amplified and converted into a single-ended signal by an adaptive amplifier, and the three frequency signals are separated and demodulated by a three-way sixth-order bandpass filter circuit. Lastly, the three frequency signals are multiplied separately for greater accuracy. The multiplied signals are further processed in the field programmable gate array (FPGA) on the digital board.





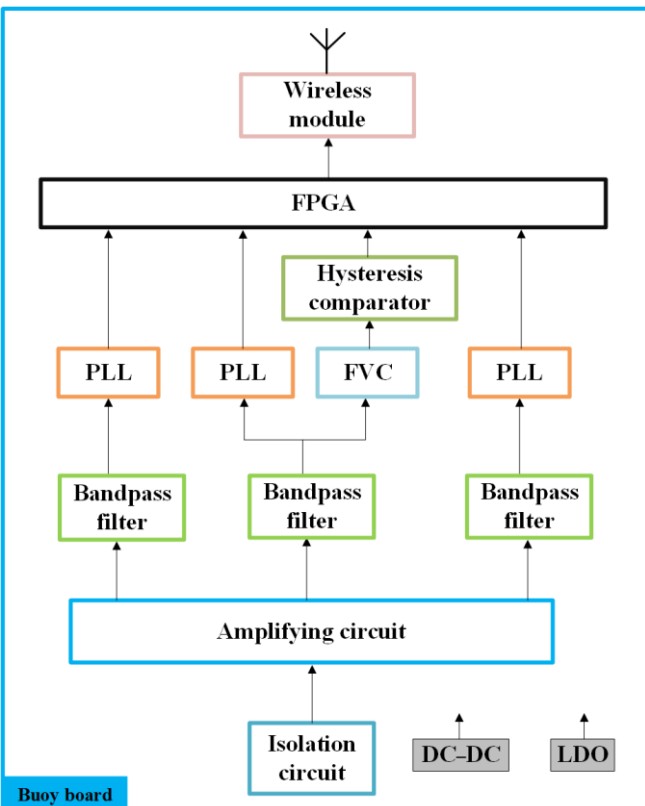

**Figure 5. Block diagram of the buoy board structure.**

The frequency multiplication circuit adopts a typical CD4046 phase-locked loop (PLL) with a counter, as shown in Figure 6. The binary counter CD4040 is used for frequency division, that is, the frequency of the output signal of the phase-locked loop VCO is $2^n$ times the frequency of the input signal. Because the maximum frequency division of counter CD4040 can be as high as $2^{12} = 4096$, the above multiplication circuit is therefore capable of multiplying the input frequency by a maximum of 4096. To enhance the measurement accuracy of XCP, the temperature and electrode coil signals will be multiplied 256 and 32 times, respectively, according to the actual situation to achieve horizontal and vertical resolutions of 1 and 0.3 m/s. Tests showed that the actual phase-locked loop frequency multiplication circuit is stable and reliable in operation. Figure 6 shows the schematic diagram of the frequency multiplication circuit. Figures 7 (**a**) and (**b**) show the output waveforms of the multiplied frequencies at 32 and 256 times, respectively, when the 1 KHz signal is input into the phase-locked loop. Figure 7 (**c**) shows the amplified waveform of the multiplied frequency at 256 times. The desired multiplication output can be obtained by varying the value of the division frequency of the counter. This phase-locked loop circuit can realize synchronous sampling and equal interval sampling well, as it improves the calculation accuracy, reduces the MCU computing time, and improves the real-time control performance (Murtianta, 2016; Zhihong, 2008).



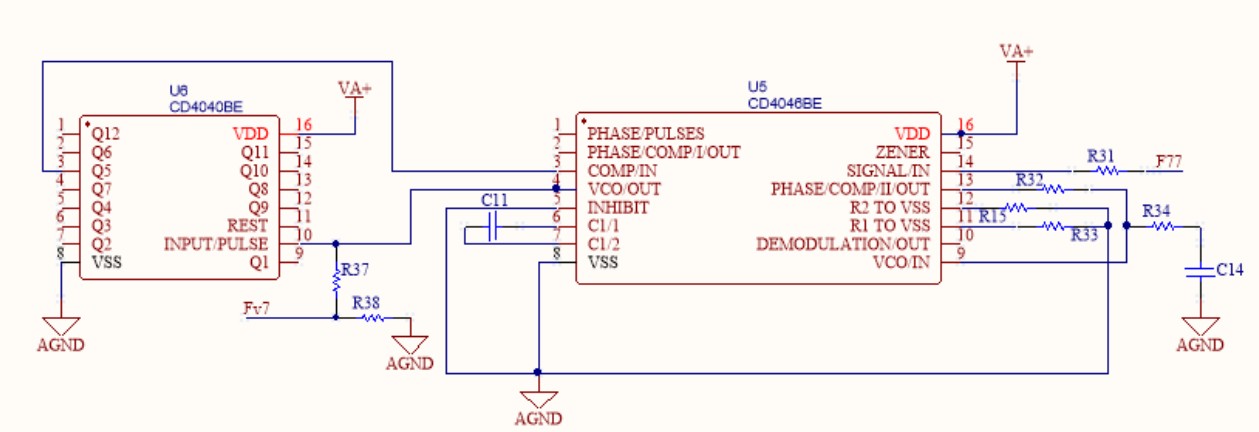

**Figure 6. Schematic diagram of the frequency multiplication circuit.**

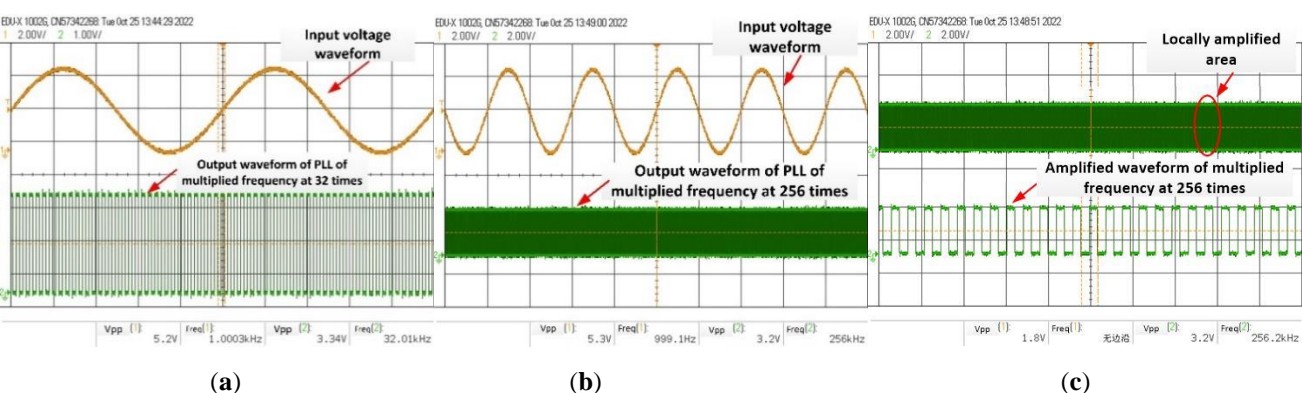

(a)                              (b)                              (c)

**Figure 7. The output waveform of a phase-locked loop of multiplied frequency at (a) 32 and (b) 256 times, and (c) amplified waveform of multiplied frequency at 256 times.**

The digital board includes the FPGA minimum system, power conversion module, water entry power module, probe releasing module, and more. The information output of the electric field, coil, temperature, and more from the main control chip after calculation, calibration, and other steps are sent to the deck unit through the wireless module.

The probe release section is controlled by a typical MOS switch circuit with a high-power 510 Ω chip resistor on the circuit board. The resistor is heated up by adding 5 V to both ends, thereby melting the resistor wire that controls the bottom cover of the probe, and after the wire is broken, the bottom cover comes off and the probe drops. The resistor heating time was tested to minimize battery power consumption, and it was found that ten seconds was enough for the wire to fuse. Therefore, the MOS tube turns off after ten seconds, and the resistor stops heating up.

The wireless data transmission module, which is the same as the one inside the buoy board, receives data at the deck unit end and connects to the PC via USB. The wireless SRWF−1028 module from Shanghai Sunray Technology Co., Ltd was used. This general-purpose transparent transmission module has low power consumption and transmitting and receiving currents at approximately 350−500 mA and 32−38 mA, respectively. It can accommodate standard or non-standard user protocols, has





excellent anti-interference capability, and has a long transmission distance with a handheld and line-of-sight distance of up to 2500 m. The baud rate is set at 9600 bps and the working frequency at 433 MHz.

### 3.3 Software design

155   Since higher-frequency signals are to be acquired, the digital master control chip of the lower part of the buoy board adopts the Cyclone series FPGA from the company Altera (Liu, 2017). The Nios II embedded processor control code, written in the C language, processes the transmitted signal data, transmits the data, times the probe release, and more. The software workflow diagram is shown in Figure 8. As the operation of the ship will interfere with wireless communication, after we release the buoy, the buoy will have a timing of 180 seconds before releasing the probe, as shown in Figure 8. As the expendable instruments are operated in a non-stop way, the ship has gone far within 180 seconds, which will not affect the wireless data

160   transmission. As mentioned in the previous section, our wireless transmission distance can reach 2500 meters, and the ship can not travel beyond 2500 meters within 180 seconds. Therefore, the problem of hull interference with wireless communication is ingeniously solved.

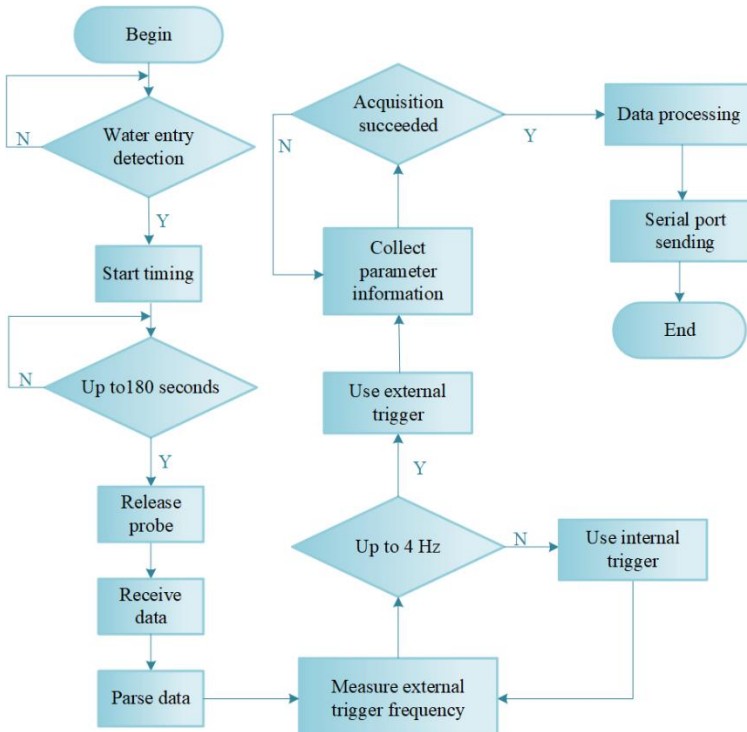

**Figure 8. Software workflow diagram of the buoy board.**

165   The upper computer software is developed in the C# language with Visual Studio 2022, and the functions include receiving data from the serial port, displaying waveform, exporting files, and so on. It has a user-friendly human-computer interaction





interface and integrates the function of sea current flow rate processing by calling Matlab code for calculation and drawing, allowing real-time processing, and displaying sea current information.

The workflow diagram of the upper computer is shown in Figure 9, and the interface of the serial port receiving data is shown in Figure 10.

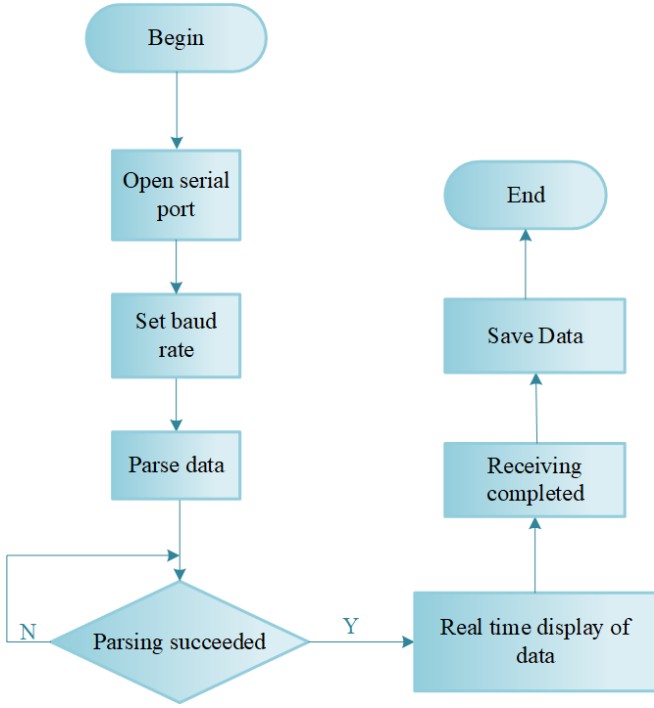

**Figure 9. Workflow diagram of the upper computer.**

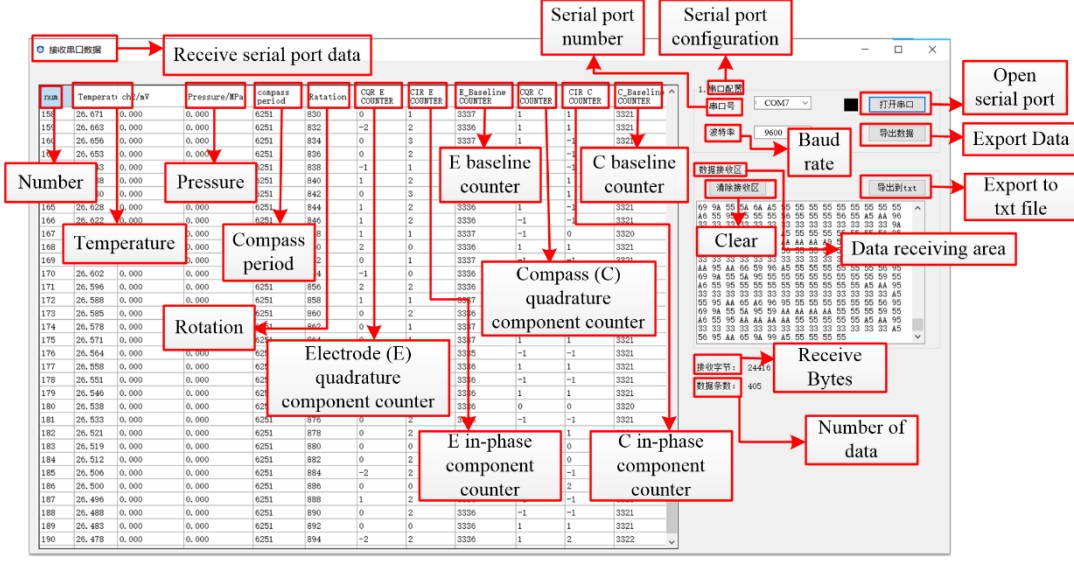

**Figure 10. The interface of the serial port receiving data.**



## 4. Results

The acquisition circuit was tested in our laboratory by simulating the entire system and comparing it with the results of the previously developed XCP.

### 4.1. Testing of the Acquisition Circuit

The accuracy of the acquisition circuit was tested. Ordinary signal generators are unable to produce nanovolt-level signals due to the relatively small electrode and coil signals, and they are also susceptible to mixed industrial frequency interference when propagated through wires. Therefore, a resistive attenuation network was added to the acquisition circuit for accuracy testing, although none will be added for the actual measurement. The attenuation circuit is shown in Figure 11, and its attenuation multiplier could be changed by varying the resistor value.

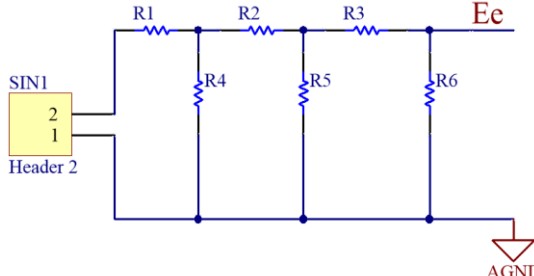

**Figure 11. Resistive attenuation network.**

Since the values of resistance, capacitance, and more will contain certain errors, to obtain accurate readings, the accurate amplification of each analogue board must be measured, that is, the device must be calibrated. We use a lock-in amplifier to calibrate the analogue boards. The attenuation multiplier was set to be approximately 3700 times, and a sinusoidal signal with a root mean square value of 0.4 mVrms at 16 Hz was output using a lock-in amplifier. The peak value of the signal was approximately 50 nV after passing through the attenuation circuit, it then entered the electrode channel, and the output signal was measured by the lock-in amplifier after amplification. The information displayed by the lock-in amplifier was used to calculate the attenuation and amplification multipliers, and this method was used to calibrate each board to determine the exact multiplier of amplification. As the coil channel had a relatively small multiplier of amplification, it was connected directly to the lock-in amplifier without attenuation. We made six analogue boards, they were numbered from one to six, and then tested for their corresponding multipliers of amplification. The measured multipliers of the electrode channel and coil channel amplifications are shown in Table 1 and Table 2, respectively. We used Excel to record and calculate the results and reserved two decimal places for the results. It can be seen from the data in the table that although each circuit board is welded with the same components, the total amplification or attenuation times are relatively large, and small differences in the values of each resistance and capacitance in the circuit lead to large differences in the amplification or attenuation times of the whole circuit board. Therefore, the calibration of each circuit board is of great significance to the accuracy of the results. Figure 12 shows





the final output waveform as seen with an oscilloscope for the coil channel of board No. 2, revealing a good signal quality with a peak-to-peak value of 3.06 V and an error accuracy that is within one thousandth of the theoretical value. There are numerous works of research on weak signal processing circuits in China, however, most of them are in respect of the fields of exploration, medicine, and biology, and the acquired signals are mostly at the microvolt level (Levkov, 1988; Si, 2016; Lai, 2012; Dohnal, 2019; Shan, 2022). In contrast, an induced electric field produced by currents with a flow rate of 1−3 cm/s at mid-latitudes was shown to have a signal of approximately 20−80 nV as measured by an electric field sensor with a spacing of 5 cm (Liu, 2012). The results of the experiments show that both the multipliers of the electrode and the coil meet the requirements for the acquisition of the multiplier of the signal.

**Table 1. Electrode channel multipliers.**

| Board No. | Signal frequency (Hz) | Attenuation multiplier | Electrode channel amplification multiplier |
|-----------|----------------------|------------------------|--------------------------------------------|
| 1 | 16 | 3766.48 | 120527.30 |
| 2 | 16 | 3735.52 | 119462.09 |
| 3 | 16 | 3769.32 | 111209.96 |
| 4 | 16 | 3732.74 | 115752.14 |
| 5 | 16 | 3756.57 | 119083.40 |
| 6 | 16 | 3769.32 | 111209.95 |

**Table 2. Coil channel amplification multiplier verification.**

| Board No. | Signal frequency (Hz) | Coil channel amplification multiplier |
|-----------|----------------------|----------------------------------------|
| 1 | 16 | 2529.23 |
| 2 | 16 | 2702.49 |
| 3 | 16 | 2604.82 |
| 4 | 16 | 2623.15 |
| 5 | 16 | 2730.33 |
| 6 | 16 | 2526.31 |



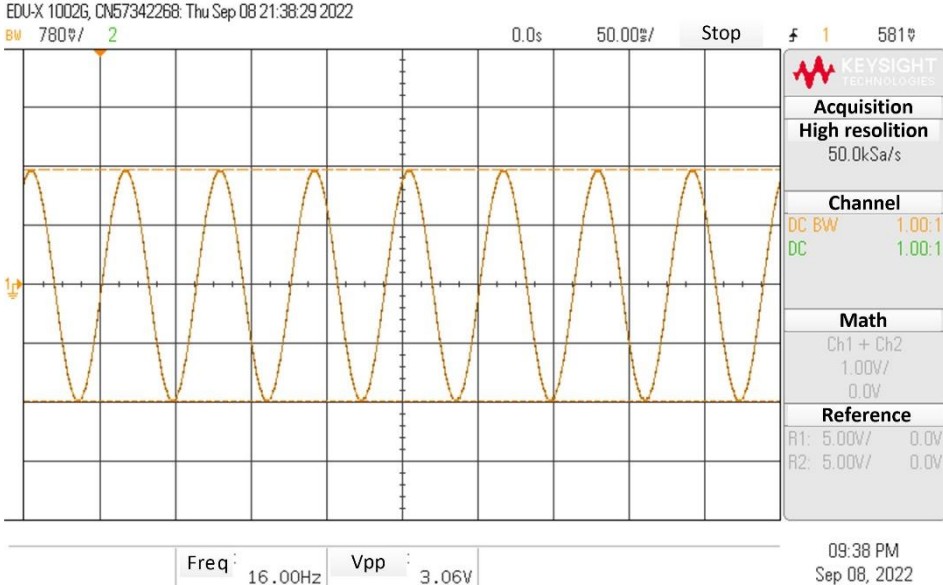

**Figure 12. The output waveform of the coil channel.**

## 4.2. Tests With Added Direct Current Field Inside and Outside the Laboratory

The laboratory test was carried out in a plastic tank with simulated seawater. In the test, because XCP is used in the ocean, we need to use seawater for the test. Because it is not convenient to obtain real seawater, we use artificial seawater instead of natural seawater indoors. The conductivity of seawater is mainly related to salinity, and the average salinity of seawater is

about 35%. Therefore, to better simulate the conductivity of seawater in the experimental environment, tap water and sea salt are configured according to the salinity of seawater. By calculating the volume of tap water in the plastic tank, determine the weight of sea salt required according to the average salinity of 35% in seawater and the salinity in sea salt. The XCP probe is required to rotate with an angular velocity of approximately 16 Hz while dropping, so that both the compass and coil signals are modulated into an approximate single-frequency signal with a frequency of 16 Hz, according to the characteristics of the

current electric field and compass coil signals. A rotating structure device was designed consisting of a small direct current brushless motor, a holder to fix the motor, an XCP probe, a fixture to fix the XCP probe, a coupling, and a conductive slip ring. The structure of the device is shown in Figure 13. The direct current motor is controlled by the controller and can rotate at a speed of up to 16 Hz. The function of the conductive slip ring is to allow for the rotation of the lower wire of the signal transmission as the motor rotates while keeping the upper wire stable and still so that the signal transmission wire will not

become tangled. The function of the coupling is to enable the XCP fixing structure to rotate coaxially by driving force from the motor shaft, which enables the XCP probe to rotate. We use a 1500-metre-long double-strand enamelled wire with an outer diameter of 0.1 mm for signal transmission, and the coupling is connected to the buoy end.





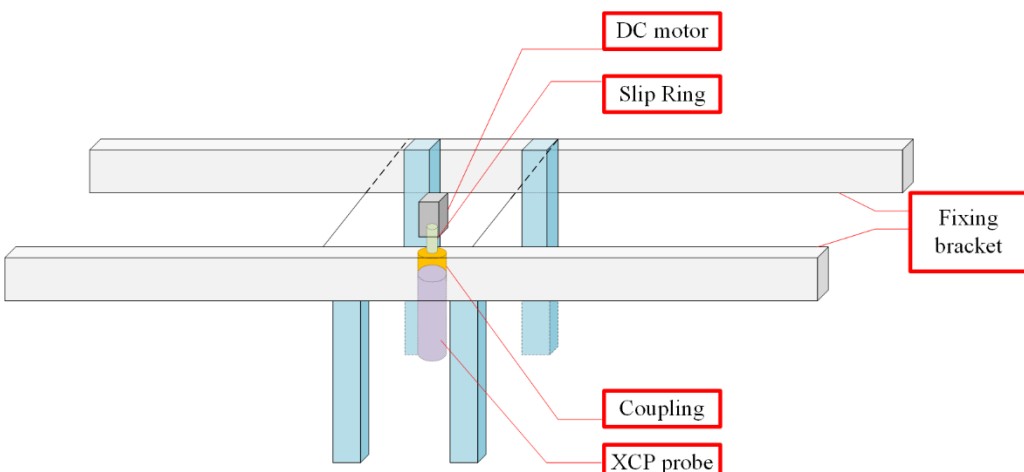

**Figure 13. Schematic diagram of rotating structure device.**

Copper plates were placed at both ends of the trough and direct current was added to both plates. By controlling the motor to rotate the XCP, the direct current was converted into alternating current, which was measured by the electrodes and transmitted inside the XCP for processing. The experimental setup is shown in Figure 14.

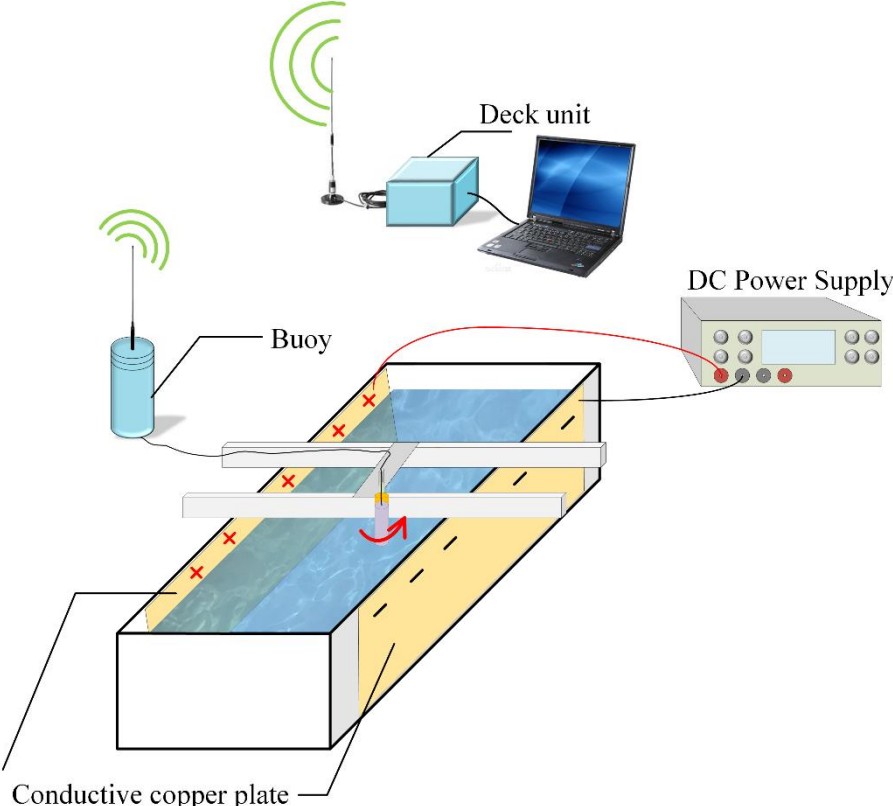

**Figure 14. Schematic diagram of the XCP experimental setup.**





The magnitude of the voltage of the direct current regulator source was altered, and the voltage data of the copper plates were measured with a digital multimeter. The data from the XCP probe was received and transmitted back to the host computer, and then processed to calculate the peak-to-peak values of the measured electrode voltages. As the experiment was conducted in the laboratory, there was a substantial amount of noise compared to the marine environment. The ocean is comparable to a low-pass filter whereby all high-frequency noise from human sources will not interfere with the ocean electromagnetic signal,

whereas a laboratory test environment may have noise interference, such as industrial frequencies. Therefore, a millivolt level signal was added to both ends of the copper plates so that the electrode measurements were set roughly to the microvolt level instead of the nanovolt level to reduce the interference from noise caused by human activity. We used the same test method to test two XCP probes. The experimental test photo is shown in Figure 15.

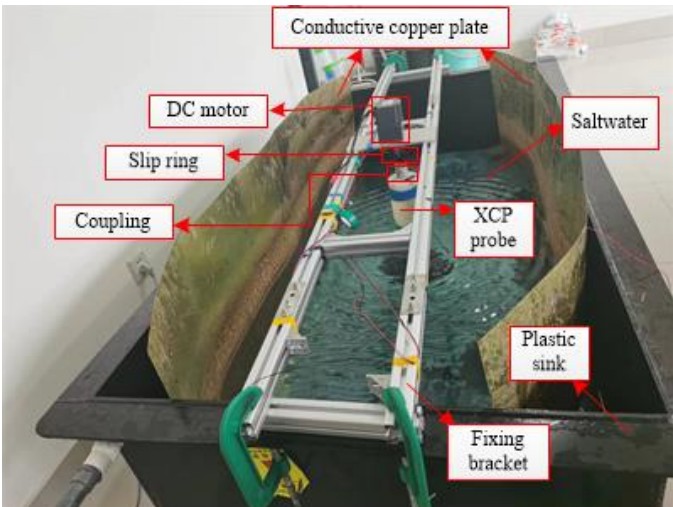

**Figure 15 Photo of the experimental environment**

A comparison of the results of the two XCP probes is shown in Figure 16. It can be seen that the measured voltages of the two probes were consistent with the trend of the copper plate supply voltage, and the results obtained by the two probes were largely the same when the copper plate supply voltage was in the approximate range of ±5mV~54mV±70mV. When the copper plate supply voltage was in the range of ±5mV~54mV±70mV, the difference between the peak-to-peak values of the electrodes

measured by the two probes was the largest at the reverse supply voltage of $30.32$ mV, which was 0.61 μV. When the copper plate voltage was less than 5 mV, the measurement results of both probes became unstable simultaneously and fluctuated in the approximate range of 2−5 μV, indicating that noise interference did exist in the laboratory environment. When the copper plate voltage was greater than 54 mV, the voltage measured by the XCP remained largely constant, indicating that the amplifier was fully saturated.






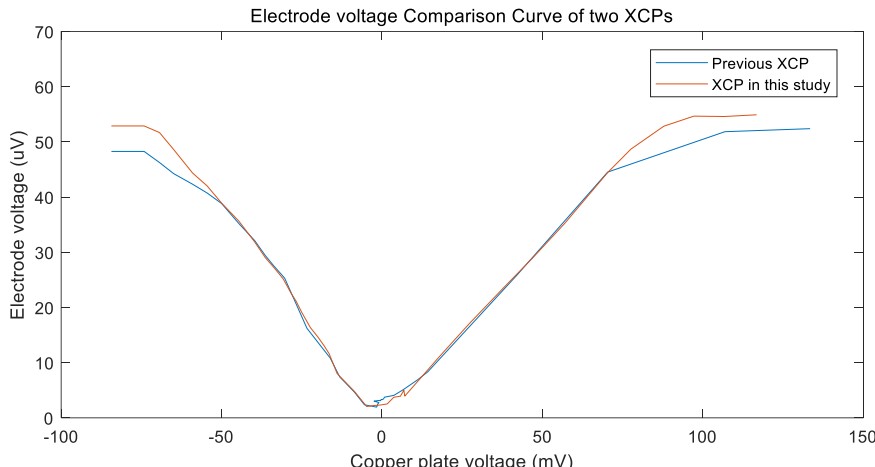

**Figure 16. Comparison of electrode voltages measured by the two types of XCPs.**

Based on the above analysis, we needed to eliminate the interference of indoor noise over the range of the results. Therefore, we selected the data in the middle of the copper plate power supply voltage between ± 15mV and ± 50mV, with the voltage remaining measured to two decimal places. We listed these voltages in Table 3 to display the data more accurately. The actual acquisition accuracy errors of the two probes are within 3.9%, and the maximum error is 0.61 μV. We surmise that the error may be related to the inherent noise in the laboratory test environment. In addition, the operation of the motor could also have an impact on the accuracy. These interferences will not occur in the marine test, because the ocean is far from polluting noise interferences such as alternating current found in the city environment. The XCP with which the comparison was made had undergone marine tests in a previous study and the results showed that this XCP worked with stability and accuracy (Liu, 2017); therefore, this result has a high reference value.

**Table 3. Comparison of test results.**

| Copperplate voltage (mV) | Electrode voltage (μV) (Previous XCP) | Electrode voltage (μV) (XCP of this study) | Delta (μV) | Error (%) |
|---|---|---|---|---|
| -49.96 | 38.82 | 38.89 | 0.07 | 0.180319 |
| -44.6 | 35.22 | 35.66 | 0.44 | 1.24929 |
| -39.70 | 32.14 | 31.89 | -0.25 | 0.77785 |
| -36.36 | 29.41 | 29.03 | -0.38 | 1.29208 |
| -33.51 | 27.22 | 27.12 | -0.1 | 0.36738 |
| -30.32 | 25.31 | 24.7 | -0.61 | 2.41011 |
| -26.68 | 20.56 | 21.06 | 0.50 | 2.431907 |
| -22.57 | 16.21 | 16.35 | 0.14 | 0.863664 |





| -16.08 | 10.81 | 11.18 | 0.37 | 3.422757 |
|--------|-------|-------|------|----------|
| 14.32 | 8.36 | 8.68 | 0.32 | 3.827751 |
| 20.50 | 12.35 | 12.75 | 0.40 | 3.238866 |
| 25.43 | 15.41 | 15.91 | 0.50 | 3.244646 |
| 30.22 | 18.46 | 18.54 | 0.08 | 0.433369 |
| 37.72 | 23.15 | 23.44 | 0.29 | 1.2527 |
| 42.28 | 26.34 | 26.46 | 0.12 | 0.455581 |
| 47.10 | 29.15 | 29.04 | -0.11 | 0.37736 |

We also compared the XCP performance with the existing XCP performance. Table 4 lists representative performance
indicators of this XCP and, for comparison, those of the previously developed XCP and the Sippican MK10A XCP (from
lockheedmartin.com). It can be seen that we successfully reduced the cost on the premise of ensuring accuracy, which is of
great significance for expendable instruments.

**Table 4. Comparison of instrument performance indicators.**

| Item | Previous XCP | XCP of this study | MK10A (Sippican) |
|------|--------------|-------------------|------------------|
| Depth (m) | 1200 | 1500 | 1500 |
| Velocity Resolution (cm/s) | 1.0cm/s | 1.0cm/s | 1.0cm/s |
| Vertical Resolution (m) | 0.3m | 0.3m | 0.3m |
| Temperature Range | 0℃ to +30℃ | -2℃ to +40℃ | 0℃ to +30℃ |
| Sampling Rate (Hz) | 16 | 16 | 16 |
| Availability | Yes | Yes | Not available in China and some certain countries |
| Approximate Cost (CNY) | 10000 | 5000 | Not available |

As for temperature, it is related to the range of VFC. Because XCP is used to measure current parameters in seawater, 0 °C is
usually a mixture of ice and water, and the temperature of the world's oceans generally varies between - 2 °C and 30 °C.
Therefore, we set the minimum threshold to - 2 °C.





## 5. Conclusions

In this study, we designed the hardware and software for a lower cost XCP. In addition, we built a test environment to test its
accuracy and stability. Specifically:

We designed the analogue circuit of the signal processing part, including the water contact switch, delayed release switch, frequency multiplication circuit, signal conditioning, and more. These circuits could also be applied to other expendable instruments and instruments that need to obtain small signals.

We conducted circuit tests. Testing revealed that the accuracy of the XCP acquisition circuit is within one thousandth of one
per cent, and the XCP analogue circuit developed for the overall system is stable and reliable with an acquisition accuracy greater than 50 nV for a 16 Hz signal, which meets the quality requirements for acquired signals of XCP instruments.

An experimental device for indoor XCP testing was introduced, and a passive source simulation experiment was carried out. The data from the passive source simulation experiments were compared with those of the previously developed XCP that was validated with marine tests (Liu, 2017). The results of the two probes were found to be highly consistent with a maximum
error of $0.61\mu V$, indicating that the XCP is stable and reliable.

The XCP developed in this study moved the processing circuit to above the surface of the water, reducing the use of underwater components and chips and effectively reducing costs. It is estimated that the cost of the XCP probe is reduced by half.

In future, we will improve accuracy, study the effect of increasing the measurement parameters and measurement distance and improve the data transmission mode. In addition, we aim to achieve mass production as soon as possible.


**Data availability:** There are no publicly available data for this study.

**Author Contributions:** Conceptualization, Q.Z. and G.C., methodology, Q.Z. and K.Z., software, Z.L. and K.Z., validation, K.Z., Z.L., P.L. and Y.L.*, formal analysis, K.Z., investigation, K.Z., resources, Q.Z., data curation, K.Z., writing—original draft preparation, K.Z., writing—review and editing, K.Z., visualization, K.Z., supervision, Q.Z., project administration, Q.Z.
and G.C., funding acquisition, Q.Z. and G.C. All authors have read and agreed to the published version of the manuscript.

**Competing interests:** The authors declare that they have no conflict of interest.

**Acknowledgements:** We would like to thank the China University of Geosciences (Beijing) and Shandong University of Science and Technology for providing a good testing environment.

**Financial support:** This research was funded by a key R&D project in Shandong, China, grant number 2019GHY112064.

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
