# Peer review of "Development of an Expendable Current Profiler Based on Modulation and Demodulation"

_Geoscientific Instrumentation, Methods and Data Systems, 2022_

## Author Response (AR1)

9-2-2023

Prof. Ralf Srama
Handling associate editor
*Geoscientific Instrumentation, Methods and Data Systems*

Dear Prof. Srama:

We wish to re-submit the manuscript titled "Development of an Expendable Current Profiler Based on Modulation and Demodulation." The manuscript ID is GI-2022-22.

We thank you and the reviewers for your thoughtful suggestions and insights. The manuscript has benefited from these insightful suggestions. I look forward to working with you and the reviewers to move this manuscript closer to publication in the *Geoscientific Instrumentation, Methods and Data Systems*.

The manuscript has been rechecked and the necessary changes have been made in accordance with the reviewers' suggestions. The responses to all comments have been prepared and attached herewith.

Thank you for your consideration. I look forward to hearing from you.

Sincerely,
Keyu Zhou, Qisheng Zhang
School of Geophysics and Information Technology
China University of Geosciences (Beijing)
Beijing 100083, China
E-mail: zqs@cugb.edu.cn

**To the esteemed Reviewers*:***

We truly appreciate the time and energy you spent in carefully reviewing our manuscript. In fact, the revision process has enriched our knowledge, and this experience will help us in our future research.

We are still keen on publishing our paper in *Geoscientific Instrumentation, Methods and Data Systems.* Therefore, we have made the necessary revisions as per your recommendations. In addition to making the corresponding revisions in the revised manuscript, we have provided our responses to your comments below.

**Reviewer #1**

1. Please use the capital letters V for voltage and check all figures.

**Answer:**

Thank you so much for your careful check, we apologize for the format problems and we revised that.

2. Please insert a space between the numeral and unit, such as mV, $\mu$V, and so on.

**Answer:**

Thank you so much for your careful check, we apologize for the format problems and we revised that.

3. Do you investigate how the operation of the ship affect wireless communication?

**Answer:**

We are grateful for your suggestion. The ship definitely affects wireless communication, and we tried to investigate it but it would take very complicated calculation, which will cause the software to run slowly and prone to errors, so we think the best way is to avoid this impact. As a result, on line 157 on page 9, we can see:

'As the operation of the ship will interfere with wireless communication, after we release the buoy, the buoy will have a timing of 180 seconds before releasing the probe, as shown in Figure 8. As the expendable instruments are operated in a non-stop way, the ship has gone far within 180 seconds, which will not affect the wireless data transmission. As mentioned in the previous section, our wireless transmission distance can reach 2500 meters, and the ship can not travel beyond 2500 meters within 180 seconds. Therefore, the problem of hull interference with wireless communication is ingeniously solved.'

That is how we solved that problem. This is the best solution we can get after considering performance, method and cost.

4. Calibration is introduced in this paper. Each XCP needs to be calibrated one by one before use, which is a bit time-consuming. Is there a simple way to deal with it?

**Answer:**

Thank you for pointing out the calibration related problems. We are sorry that because every component is not exactly the same, such as resistors, capacitors and chips, there are certain differences in resistance, capacitance and performance. Although this difference may be very small, it does exist. Moreover, our magnification is relatively large, which may amplify these

differences. Therefore, in order to ensure the accuracy of the measurement results, we need to accurately calibrate each XCP before leaving the factory, and write the parameters on the package of the product for the user to view.

5.  I recommend enhancing the Conclusions by mentioning limitations of the instrument developed in your study or potential barriers to its use.
    **Answer:**
    We appreciate raising this point. We added the shortcomings of the XCP developed in this paper.

**Supplementary content:**

However, there are still remain some areas wherein we can continue furtherto optimize and improve., fFor example, our accuracy remains the same as before, and the measurement parameters have not been increased. In the future, we will improve the accuracy, study the effect of increasing the measurement parameters and measurement distance, and improve the data transmission mode. In addition, we aim to achieve mass production as soon as possible.

**Reviewer #2**

1、 The XCP developed in this paper saves cost by reducing the main control chip of probe and buoy. However, because XCP is disposable, it does not require much performance and long-term reliability, so the price of the main control chip is also low. At the same time, the XCP adds modulation and demodulation circuits and hardware, so does the XCP solve the problem of high cost?

**Answer:** We appreciate the reviewer for raising this point. XCP measured the induced electric field generated by the current cutting the geomagnetic field, which is a considerably small signal at a nanovolt level; therefore, accurate current information needs to be measured, certain requirements must be satisfied for the accuracy and stability of the instrument. The measurement adopts the method of measuring frequency, which needs to generate frequency signals and to simultaneously compare and measure the frequencies and phases of multiple signals, including numerous calculations. This would require a large amount of resources; therefore, it is necessary to use the main control chip considering certain resources and performance, which also determines the price of the main control chip. The added modulation and demodulation component of XCP is mainly composed of amplifiers and resistance-capacitors. In this design, four operational amplifiers are used, and four amplifiers are integrated on one chip; therefore, the number of chips added is small, that is, only two chips are added. In addition, the modulation and demodulation component needs counters. As shown in Figure 6 in the paper, there are two types of chips (CD4046 and CD4040), which are extremely low in cost. In addition, we have replaced and optimized some other components of the entire system and selected the scheme with the best cost performance. Therefore, the cost of the system is low.

Following careful considering of your suggestion, we revised the relevant information to Chapter 1 of the manuscript. To make it easier for you to review the added content, the relevant information is given below.

**Supplementary content:**

Moreover, the digital circuitry inside the underwater probe was removed from our design. Because XCP measured the induced electric field generated by the current cutting the geomagnetic field, which is a considerably small signal at a nanovolt level, if accurate current information needs to be measured, certain requirements must be satisfied for the accuracy and stability of the instrument. The measurement adopts the method of measuring frequency, which needs to generate frequency signals and to simultaneously compare and measure the frequencies and phases of multiple signals, including numerous calculations. Therefore, the measurement requires significant resources; therefore, the main control chip should have certain resources and performance, which determines the price of the chip. In this design, the added modulation and demodulation component of XCP is mainly composed of amplifiers, counters, resistors and capacitors, and the cost of this chip is low. In addition, we replaced and optimized some other components of the entire system and selected the scheme with the best cost performance.

2、Only the comparison between the studied XCP and the previously developed XCP is carried out in the laboratory, it is recommended to compare it with mature product to better reflect the performance of the instrument.

**Answer:** Your suggestion is highly valued, and we express our gratitude for helping us improve our work. The performance of previously developed XCP in the laboratory has been proven. The previous XCP developed in the laboratory has gone through many ocean experiments and has been compared with other mature products (Doppler velocimeter). The results indicated that the XCP has stable performance and performs accurate measurement, which has been reflected in a previous study (Liu, 2017). Therefore, the previous XCP developed in the laboratory has a certain reference value. To better verify the performance of XCP considered in this study, we carried out further experiments comparing with a Nortek Vectrino Profiler Acoustic Doppler Velocimetry according to your suggestions and presented the experimental results in the paper.

Following careful considering of your suggestion, we revised the relevant information to Chapter 1 of the manuscript. To make it easier for you to review the added content, the relevant information is given below.

**Supplementary content:**

In addition, we compared it with mature products to better reflect the performance of the instrument. We carried out an XCP probe rotation simulation test under the hydrodynamic large-scale velocity environment at the Beihai Environmental Monitoring Center of the State Oceanic Administration of China. The key laboratory of Bohai ecological early warning, protection and restoration of the center has built a large hydrodynamic wave current simulation tank, covering an area of approximately 125 m2, with a length of 32.0 m, internal width of 0.8 m, internal depth of 2 m, and maximum working depth of 1.5 m. The working medium uses seawater. When the working water depth is 1.5 m, it can produce a uniform and stable flow field adjustable within 0.1 m/s, which satisfies the requirements of this experiment. In this experiment, Nortek Vectrino Profiler Acoustic Doppler Velocimetry was used as the comparison equipment. The specific layout of the experimental equipment is shown in Figure 19.

[Figure]

Figure 19. Photograph of ADV and XCP comparison experiment

Under different current-generation gears of the flow rate tank (based on the working current level of the current generation generator: 8 A, 9 A, 10 A, 11 A, and 13 A), we performed the comparison test of ADV and XCP rotating flow rate. Owing to the limited accuracy of the flume, the principle of the two test equipment is different, and the accuracy and error sources are different; therefore, some errors would be present. As shown in Figure 20, the overall change trend of the velocity data curve collected by ADV and XCP is consistent; as the current increases, the measured flow rate increases. Moreover, the velocity value measured using the acoustic method is slightly higher than that using the electromagnetic method, which may be caused by the slight interference of the metal reinforcement framework of the tank and the current generator with the geomagnetic field, which does not exist in the actual marine application. The maximum velocity error occurs at the ninth speed position of the current generator, with a velocity deviation of 0.0521 cm/s. For the XCP measurement accuracy of 1 cm/s, the deviation is approximately 5.2% of the total accuracy, which satisfies the XCP test requirements within the allowable range of test error.

[Figure]

Figure 20. Comparison of ADV and XCP velocity

3、 The resistive attenuation network is added at the signal input side, and the thermal noise of the resistance itself may also affect the measurement accuracy of the input signal, whether it is considered.

**Answer:** Thank you for highlighting the thermal noise-related problems. In fact, we have considered the noise caused by the resistance of the attenuation circuit. The resistance thermal noise voltage is proportional to the resistance value, bandwidth, and square root of the temperature (Kelvin). In essence, the resistance thermal noise is unavoidable. However, to minimize the noise caused by resistance, we should avoid selecting resistors with large resistance values. Through

actual measurement and experiment, we found that the influence of resistance thermal noise can be ignored. In addition, the reason for designing the attenuation circuit is that an instrument that can generate small signals at nanovolt level is not available in the laboratory, and the attenuation network would only be used in the calibration and measurement in the laboratory. In the actual ocean test, originally, the signal is generated at the level of nanovolts; therefore, the attenuation network is not required in the actual ocean application.

Following careful considering of your suggestion, we revised the relevant information to Chapter 1 of the manuscript. To make it easier for you to review the added content, the relevant information is given below.

**Supplementary content:**

The resistance thermal noise voltage is proportional to the resistance value, bandwidth, and square root of the temperature (K). Essentially, the resistance thermal noise is unavoidable. However, to minimize the noise caused by resistance, selecting resistors with large resistance values should be avoided.